# Pupillary reflex and behavioral masking responses to light as functional measures of retinal degeneration in mice

Ethan O. Contreras[1,2], Carley G. Dearing[1,2,3], Crystal A. Ashinhurst[1,2], Betty A. Fish[1,2], Sajila N. Hossain[1,2], Ariana M. Rey[1,2], Primal D. Silva[1,2], Stewart Thompson[1,2]*

**1** Department of Psychology, New Mexico Tech, Socorro, NM, United States of America, **2** Department of Biology, New Mexico Tech, Socorro, NM, United States of America, **3** College of Veterinary Medicine and Biomedical Science, Colorado State University, Fort Collins, CO, United States of America

* stewart.thompson@nmt.edu

**Data Availability Statement:** Data are available from Dryad (DOI: 10.5061/dryad.9kd51c5g5).

**Funding:** Robert Cormack Memorial award from Dr. Frank and Sheri Etscorn to ST. The funders had

## Abstract

### Background

Pre-clinical testing of retinal pathology and treatment efficacy depends on reliable and valid measures of retinal function. The electroretinogram (ERG) and tests of visual acuity are the ideal standard, but can be unmeasurable while useful vision remains. Non-image-forming responses to light such as the pupillary light reflex (PLR) are attractive surrogates. However, it is not clear how accurately such responses reflect changes in visual capability in specific disease models. The purpose of this study was to test whether measures of non-visual responses to light correlate with previously determined visual function in two photoreceptor degenerations.

### Methods

The sensitivity of masking behavior (light induced changes in running wheel activity) and the PLR were measured in 3-month-old wild-type mice (WT) with intact inner retinal circuitry, *Pde6b-rd1/rd1* mice (*rd1*) with early and rapid loss of rods and cones, and *Prph2-Rd2/Rd2* mice (*Rd2*) with a slower progressive loss of rods and cones.

### Results

In *rd1* mice, negative masking had increased sensitivity, positive masking was absent, and the sensitivity of the PLR was severely reduced. In *Rd2* mice, positive masking identified useful vision at higher light levels, but there was a limited decrease in the irradiance sensitivity of negative masking and the PLR, and the amplitude of change for both underestimated the reduction in irradiance sensitivity of image-forming vision.

### Conclusions

Together these data show that in a given disease, two responses to light can be affected in opposite ways, and that for a given response to light, the change in the response does not accurately represent the degree of pathology. However, the extent of the deficit in the PLR

no role in study design, data collection and analysis, decision to publish, or preparation of the manuscript.

**Competing interests:** The authors have declared that no competing interests exist.

means that even a limited rescue of rod/cone function might be measured by increased PLR amplitude. In addition, positive masking has the potential to measure effective treatment in both models by restoring responses or shifting thresholds to lower irradiances.

## Background and approach

Animal models are widely used in studies of inherited retinal diseases, for understanding pathology and then for testing safety and efficacy of emerging treatments [1, 2]. Acuity, contrast sensitivity, and the luminance range of visual behavior provide direct measures of visual function [3–8]. However, these tests can lack sensitivity when vision is severely reduced, and meaningful gains or losses in vision may be beyond this sensitivity limit. As a result, the non-image-forming responses to light have increasingly been used as a surrogate for visual function [9]. For example, there has been some success in implying rod and cone photoreceptor function from the PLR, particularly by contrasting responses to blue and red light [10–13].

An alternative measure of retinal function should ideally be sensitive to a very limited degree of function, provide objective and quantitative measurement, and meaningfully represent vision. In mice, non-image-forming responses to light include: (1) circadian rhythm entrainment that aligns internal clock time with external solar time; (2) modification of sleep propensity; (3) suppression of pineal melatonin synthesis; (4) the PLR that adjusts pupil size to changing light conditions; (5) 'negative masking' which is a suppression of locomotor activity in bright light; and (6) transient increases in activity at lights-on [9, 14–18].

These responses are driven by intrinsically photosensitive retinal ganglion cells (ipRGCs) which have a response to light that is modified by rod/cone input [9]. This means that two distinct retinal diseases can have very different effects on a specific non-image-forming response [19]. In addition, there are subtypes of ipRGC with differences in intrinsic response to light and in rod/cone input that support differences in kinetics and irradiance/spectral sensitivity of specific responses [20–23]. This response-specialization means a disease can have distinct effects on two different non-image-forming responses, and the most informative responses for one retinal disease may not be suited to other diseases.

The PLR and negative masking are typically the non-image-forming responses best suited to measurement of disease progression because they allow resampling every few days, although recently identified transient responses to lights-on may also prove useful [16, 24, 25]. Additionally, negative masking protocols have the potential to also identify the presence of image-forming-vision in dim light, termed positive masking because it describes an increase in wheel running over baseline in complete darkness [7]. The goal of this study, was to determine the capacity of the PLR and masking behavior for measuring retinal function in different forms of rod/cone photoreceptor degeneration in mice.

For contrasting rod/cone degenerations, in this study we compared two forms of inherited retinal degeneration in mice *Mus Musculus*, the C3H/HeJ-*Pde6b*$^{rd1}$ mouse (*rd1*) and the C3A.Cg-*Pde6b*$^+$ *Prph2*$^{Rd2}$/J mouse (*Rd2*) [26]. Both *rd1* and *Rd2* mice suffer progressive loss of rod and cone photoreceptor cells, but with markedly different time course. Mice homozygous for the *rd1* allele of cGMP phosphodiesterase 6b (Pde6b MGI:97525) undergo rapid degeneration of rods and cones, with rod-cone generated light evoked responses undetectable by P28 using direct recording of the retina by multi-electrode array, so dysfunction overlaps with retinal development [26, 27]. In mice homozygous for the *Rd2* allele of Peripherin 2 (*Prph2* MGI:102791), the rod/cone outer segment fails to form and rods and cones progressively degenerate, but at 3 months photoreceptors are responsive to bright light and support useful vision in bright light [8, 28, 29].

## Methods

Approach and all methods were based on our previous experience with these techniques, animal models of retinal degeneration, and analysis methods. There were no conflicts of interest. Data is publicly available at Dryad, doi:10.5061/dryad.9kd51c5g5.

This study was carried out in strict accordance with the recommendations in the Guide for the Care and Use of Laboratory Animals of the National Institutes of Health. Use of animals in research was approved by the New Mexico Tech Institutional Animal Care and Use Committee. All procedures were non-invasive and mice were lightly sedated when restraint stress would affect measurements. Total number of animals used in this study was 64.

To constrain variables, mice of the same age and C3H genetic background were studied under the same conditions. *rd1*, *Rd2* and wild-type control mice (C3Sn.BLiA-*Pde6b*<sup>+</sup>/DnJ) were sourced from the Jackson Laboratories (Bar Harbor, ME) and then bred on site. Mice were maintained in a repeating cycle of 12-hours fluorescent white light at $\sim20\mu Wcm^{-2}s^{-1}$, then 12-hours dark, except when undergoing experiments. Food and water were available *ad libitum*. Light measurements were made using a PM103 power meter (Macam Photometrics Ltd, Livingston, UK). There were no humane end-points in this study.

*Retinal histopathology* was assessed in Wild-type, *rd1* and *Rd2* mice according to previously described protocols [19]. Mice were those previously used in PLR experiments. Sample size (n = 3 per genotype. 1 male and 2 female per group) was small because this was simply to confirm multiple previous descriptions of retinal anatomy in these strains [8, 19, 26, 30]. Mice were humanely euthanized by anesthetic overdose for 5-minutes followed by cervical dislocation. Eyes were collected and fixed in 4% paraformaldehyde (PFA) (Sigma-Aldrich, St. Louis, MO) for 4 hours, then transferred to 1% phosphate buffered saline (PBS) and stored at 4˚C. After removal of the lens, eyes were infiltrated and embedded in acrylamide solution, then embedded in Tissue Freezing Medium (General Data, Cincinnati, OH) and sectioned at 7μm along the anterior-posterior axis on a Shandon FE Cryostat (Thermo Fisher scientific, Waltham, MA). Sections were stained with hematoxylin-eosin and images recorded on a Leica ICC50 HD. This experiment simply confirmed retinal anatomy, so genotype was not masked.

*The electroretinogram* was recorded in Wild-type (n = 6), *rd1* (n = 8), and *Rd2* mice (n = 8) according to previously described protocols [8]. Mice were those previously used in PLR experiments, with equal numbers of male and female animals in each group. Sample size was based on multiple previous descriptions of retinal function in these strains [30–32]. Mice were dark-adapted then sedated with ketamine:xylazine (100:10 mg/kg). Pupils were dilated using Tropicamide 1% (Falcon Pharmaceuticals, Fort Worth, TX) and corneas moistened using GenTeal (Novartis, East Hannover, NJ). Animals were positioned on a temperature-regulated platform and electrodes were placed for corneal contact, midline subdermal reference, and ground. Five flashes of 4-milliseconds at 25cd.s.m$^2$ were applied with an inter-stimulus interval of 60 seconds and responses were recorded using an Espion E2 system (Diagnosys LLC, Lowell, MA). Response amplitude was compared by Mann-Whitney test in Prism (Graphpad, San Diego, CA). This experiment simply confirmed retinal function, so genotype was not masked.

*Negative masking responses* were assessed in Wild-type, *rd1*, and *Rd2* mice according to previously described protocols [7]. Sample size was based on group size sufficient for statistical power in other retinal diseases (n = 12 for each genotype, and first test at post-natal day 90 for all mice) [7, 24, 33]. Only male mice were tested because estrus increases variability in baseline activity levels. Mice were individually housed in wheel cages (Harvard Apparatus, Holliston, MA) mounted in custom built environmental control cabinets. Wheel running was continuously recorded using a customized ClockLab data acquisition system (Actimetrics, Inc. Evanston, IL). Between tests, mice were maintained under a cycle of 12-hours fluorescent white light

at ~20μWcm$^{-2}$s$^{-1}$, then 12-hours dark. Animals were allowed to acclimatize to wheel cages for 14 days prior to testing. The testing schedule was a 3-day repeating experimental cycle of pre-pulse baseline day, pulse day, and maintenance day. On pulse days, starting one hour after daily dark onset, light at a defined irradiance was applied to mice in their home cage for 1-hour. Under this protocol, circadian activity remains entrained throughout testing, but acute changes in activity during the dark phase are induced. Cinegel Neutral density film (Rosco, Stamford, CT) was used to regulate the irradiance of the applied light. Six light levels over 5-log units of irradiance were applied in a non-sequential order that distributed bright and dim pulses over the course of testing: 0.002, 0.02, 1.55, 0.18, 34.1 then 15.3μWcm$^{-2}$s$^{-1}$. Changes in activity over the 1-hour light treatment were calculated as % of baseline activity at the corresponding time on the preceding day for each animal. Genotype was masked for experimenter analyzing the data using non-identifying sequential numbers in data acquisition file names. Data was grouped, and distribution of data was tested using the D'Agostino-Pearson normality test in Prism (GraphPad). Variable slope sigmoid dose response curves were fitted to data in Prism (Graph-Pad). The irradiance producing a half maximal response (EC50) and hill-slope were calculated in Prism from fitted curves, with a fixed constant for the minimum set at 0%. To avoid distortion of EC50 calculation by positive masking (activity greater than baseline), a fixed maximum was set at 100% for EC50 derivation only. Features of fitted curves were then compared by an F-test of a two-fit comparison in Prism. Curves were fitted to the data sets for two genotypes independently and then to the combined data set for both genotypes; the effect of combining data sets on the quality of fit to a given parameter was then used to calculate if there was a difference.

*Pupillary light reflexes* were tested using a chromatic contrast approach as previously described [34]. This approach exploits differences in efficiency of photopigment activation at different wavelengths. Opsin photopigments have a characteristic absorption spectrum but the wavelength that most efficiently activates a mouse photopigment (λmax) is specific to that pigment: Short-Wavelength-Sensitive (SWS) opsin = 360nm, Melanopsin = 480nm, Rod-opsin = 498nm, and Medium-Wavelength-Sensitive (MWS) opsin = 508nm. Relative quantum efficiency for mouse photopigments corrected for lens-absorption then normalized to 1 is: 480nm SWS = 0.00001, Melanopsin = 1.0, Rod-opsin = 0.92, and MWS = 0.83; 622nm SWS <0.00001, Melanopsin = 0.0006, Rod-opsin = 0.005, and MWS = 0.013 [35]. Although the percentage of photons that will activate an opsin is much lower for all mouse photopigments at 622nm, it is higher for Rod-opsin (0.5%), and MWS-opsin (1.6%) than it is for Melanopsin (0.06%). This means the 622nm PLR can be largely attributed to the rod and MWS-cone.

The chromatic PLR was measured in wild-type (n = 9, 4 male, 5 female), *Rd2* (*n* = 10, 5 male, 5 female), and *rd1* (*n* = 9, 4 male, 5 female) mice according to previously described protocols [34]. Sample size was based on group size sufficient for statistical power in other retinal diseases [11, 25, 34, 36, 37]. Briefly, mice were dark adapted for a minimum of 2-hours then lightly sedated using 46mg/kg ketamine with 4.6mg/kg of xylazine. Sedation avoids the stress-induced pupil dilation of awake restraint, and therefore allows measurement of pupil responses at low light levels. Using infra-red cameras, animals were positioned on a test platform and pupil responses recorded from both eyes using A2000 pupillometer (Neuroptics, Laguna Hills CA). Change in pupil size was recorded on a dark-adapted background to a 1-second red stimulus (622 ± 3nm) then a 1 second blue stimulus (480 ± 3nm), separated by a 59 second inter-stimulus interval: 1-second stimuli test rod/cone-generated pupil responses. Recordings at 0.01, 0.1, 1.0 and 10μWcm$^{-2}$s$^{-1}$ were separated by a minimum of 3 non-test days. All animals were tested at a single irradiance during any given session, with non-sequential ordering of 0.01, 1.0, 10 then 0.1μWcm$^{-2}$s$^{-1}$. Genotype was masked by Thompson using simple numbered cage cards. Recordings with poor pupil acquisition were discarded because pupil ellipse fitting prevents reliable measurement of response peak. The exclusion criteria were

determined prior to recording and identified by failed ellipse fitting during recording. Genotype group was identified after response amplitudes to the different stimuli had been calculated. Data was grouped by genotype.

Distribution of data was tested using the D'Agostino-Pearson normality test in Prism (Graph-Pad). Dark-adapted pupil size was compared by unpaired 2-tailed parametric *t*-test with Welch's correction for different Standard Deviations between genotypes. The 'Initial response' primarily generated by rod/cone activation, was defined as the maximal constriction within 2 seconds of stimulus onset. Pupil constriction was converted to percent constriction against baseline for each animal and test, then plotted as an irradiance-response function, and fitted with a sigmoidal dose response in Prism (GraphPad, San Diego, CA), using 0% pupil constriction as constrained minimum, and 95% pupil constriction as a constrained maximum, reflecting the range of pupil constriction in mice. Full dose-response curves were not established so comparison was by Welch's *t*-test of response amplitude at the highest irradiance for red and for blue stimuli.

## Results

### Retinal anatomy and function

Degree and form of retinal degeneration was consistent with previous descriptions of C3Sn wildtype, C3H *rd1*, and C3 *Rd2* mice (**Fig 1**). Wild-type mice showed normal outer and inner retina and robust ERG a- and b-waves. *rd1* mice showed essentially complete loss of the outer retina and an unrecordable ERG consistent with loss of the rod/cone generated response to light. *Rd2* mice showed partial loss of the outer retina and a small but recordable ERG b-wave demonstrating some photoreceptor generated response to light.

### Negative masking of running wheel activity

Baseline wheel running activtiy was not different between wild-type (wheel revolutions between Zeitgeber time 14 and 15, Mean ± SD = 294.0 ± 25.1) and *rd1* mice
(Mean ± SD = 274.3 ± 64.5; Two tailed unequal variance t-test P = 0.33; n = 12), or between wild-type and *Rd2* mice (Mean ± SD = 311.4 ± 79.0; P = 0.48; n = 12).

In wild-type mice, running wheel activity was suppressed by light in a dose dependent manner (**Fig 2**). Activity in dim light was greater than the baseline, which is attributable to the effect of useful vision on running wheel use. The irradiance producing a half maximal response (EC50) was 0.54 $\mu Wcm^{-2}s^{-1}$, with a slope of 0.69.

In *rd1* mice, there was no augmentation of running wheel activity in dim light, consistent with profound blindness. Negative masking was induced at significantly lower irradiances (EC50 0.04 $\mu Wcm^{-2}s^{-1}$, F-test $P < 0.001$, F = 14.2; n = 12). The slope of the response was also significantly increased in *rd1* (1.81; F-test $P < 0.005$, F = 10.3; n = 12), suggesting a change in the quantum efficiency: this would be consistent with a change in photoreceptor contribution to the response with the complete loss of rod and cone photoreceptors.

In *Rd2* mice, running wheel activity was augmented at irradiances below the threshold for negative masking, indicating useful vision is retained at those light levels. Negative masking was induced at significantly higher irradiances (EC50 6.69 $\mu Wcm^{-2}s^{-1}$; F-test of EC50 $P < 0.0001$, F = 31.1; n = 12). The slope of the response was not different from wild-type (0.80; F-test of slope $P < 0.60$, F = 0.28; n = 12), suggesting a retained rod and cone contribution.

### Pupillary light reflex

Individual pupil traces for the brightest stimulus (10.0 $\mu Wcm^{-2}s^{-1}$) show that all mice had a functional pupil constriction to light (**Fig 3**). However, in *rd1* mice there was no pupil

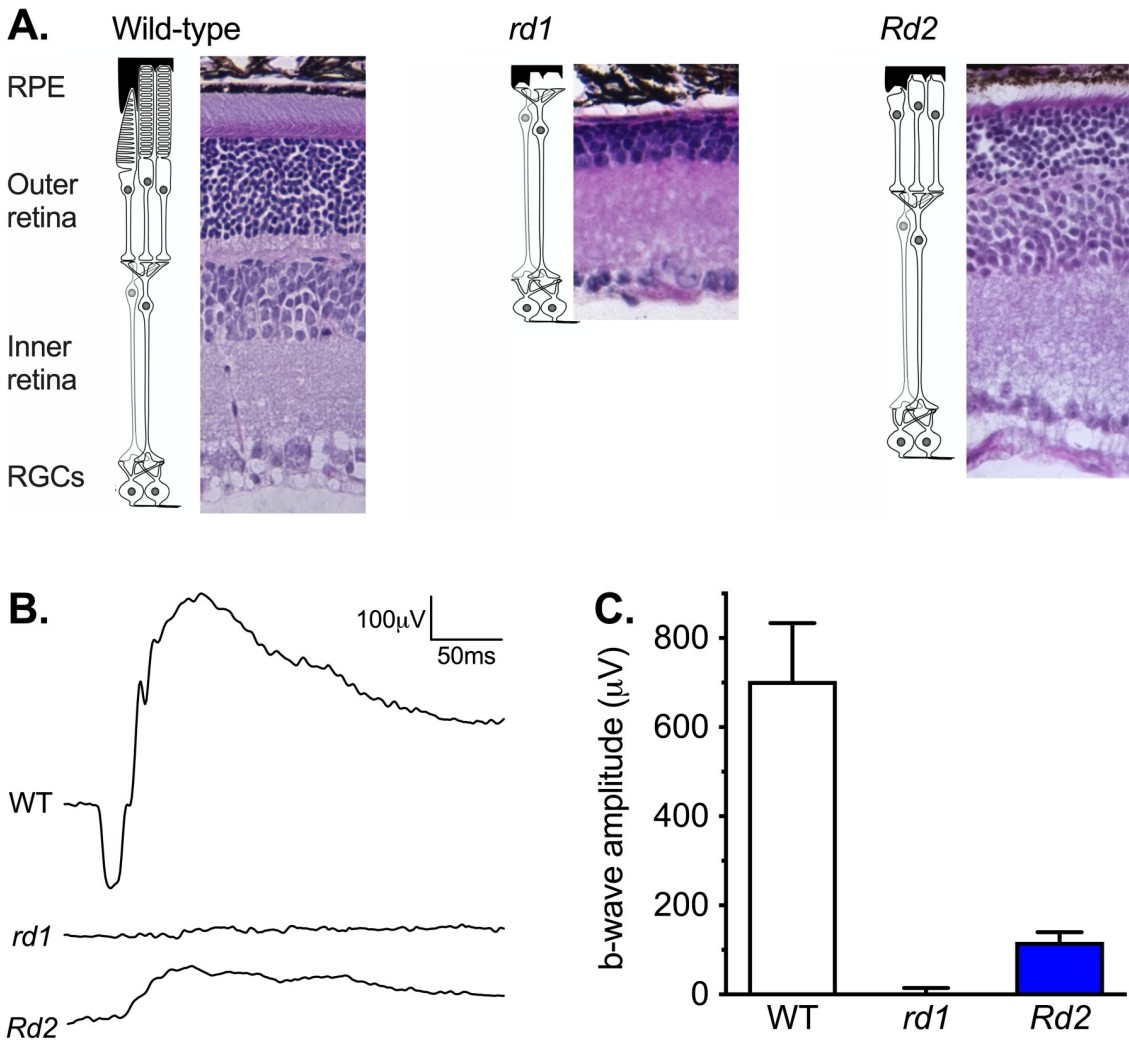

**Fig 1. Retinal anatomy and function of wild-type, *rd1* and *Rd2* mice.** (A) H&E stained sections of retina from wild-type, *rd1*, and *Rd2* mice. Gross layers of the retina are labeled: RPE = retina pigment epithelium; the outer retina with the rod and cone photoreceptor cells; the inner retina with bipolar and amacrine cells; and RGCs = the Retinal Ganglion Cell layer. (B) Electroretinogram traces in response to light from 4-millisecond 25cd.s.m$^2$ flashes of light in wild-type (n = 6), *rd1* (n = 8), and *Rd2* mice (n = 8). (C) The derived Mean ± SEM electroretinogram b-wave amplitude. Reduction in ERG b-wave was significant for both *rd1* (Mann-Whitney test $P < 0.001$; n = 6, 8) and *Rd2* mice ($P < 0.001$; n = 6, 8).

constriction to bright red light (10 µWcm$^{-2}$s$^{-1}$ at 622nm) and a pronounced reduction in the response to bright blue light (10µWcm$^{-2}$s$^{-1}$ at 480nm). In *Rd2* mice there was pupil constriction to bright red and bright blue light but amplitude was reduced for both stimuli, and with a more pronounced loss of response amplitude to red light than to blue light.

Combined traces at the range of applied irradiances show the dose dependence of the PLR (Fig 4). Comparison of responses to stimuli confirmed that the PLR was both color and irradiance dependent in wild-type, *rd1*, and *Rd2* mice. For the brightest red stimulus, constriction was significantly reduced in *rd1* (Mean ± SEM constriction for wild-type 41.3% ± 5.9; *rd1* 0.01% ± 0.48; $P < 0.0001$; t = 21.0; n = 9), and in *Rd2* (24.8% ± 7.5; $P < 0.0001$; t = 5.4; n = 9, 10). Similarly, constriction was also significantly reduced in response to a 1-second blue stimulus at 10 µWcm$^{-2}$s$^{-1}$ in *rd1* (Mean ± SEM constriction for wild-type 49.8% ± 8.4; *rd1* 13.3% ± 6.3; $P < 0.0001$; t = 10.2; n = 9), and in *Rd2* (41.3% ± 7.3; $P < 0.05$; t = 2.3; n = 9, 10).

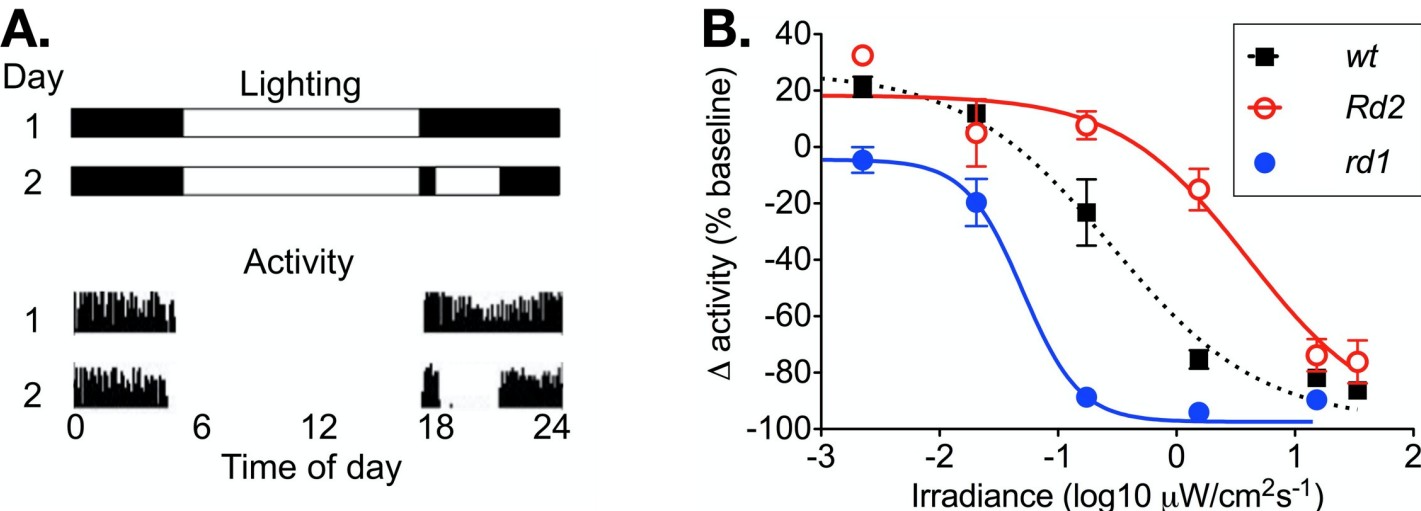

**Fig 2. Masking of activity by light.** (A) A representation of the lighting schedule for pre-test baseline day (1) and test day (2) are shown above wheel running activity from baseline (1) and test days (2). (B) The relationship between irradiance and change in wheel running activity is shown over a 4-log unit range of irradiance for wild-type (n = 12), *rd1* (n = 12), and *Rd2* mice (n = 12). Mean ± SEM activity at each irradiance is calculated as a percentage of baseline at 0%, which is determined from activity at the same time on the pre-test day. Variable slope sigmoid dose response curves are fitted to data.

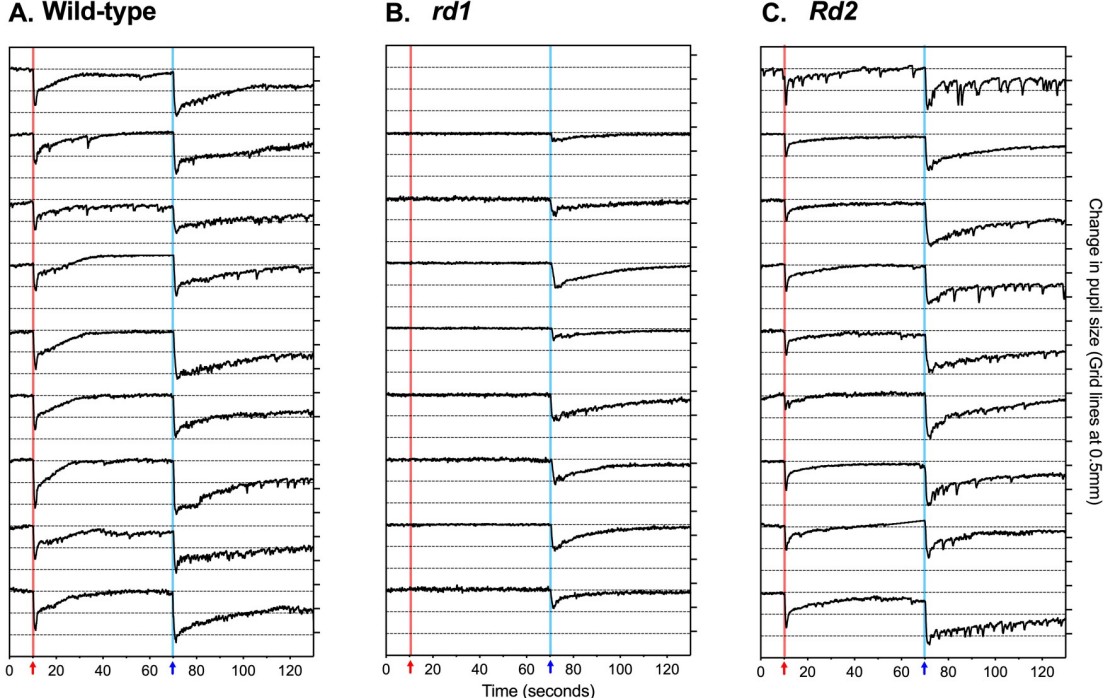

**Fig 3. Individual traces of change in pupil size to 1-second 10.0 μWcm$^{-2}$s$^{-1}$ stimuli.** Panels show individual traces of pupil size over time for (A) wild-type (n = 9), (B) *rd1* (n = 9), and (C) *Rd2* mice (n = 10): only successfully recorded traces are included. Change in pupil size is shown in mm, with grid lines showing 0.5mm intervals. The 1-second red stimulus at 10-seconds is shown by a red arrow and red background line. The 1-second blue stimuli at 70 seconds is shown by a blue arrow and blue background line.

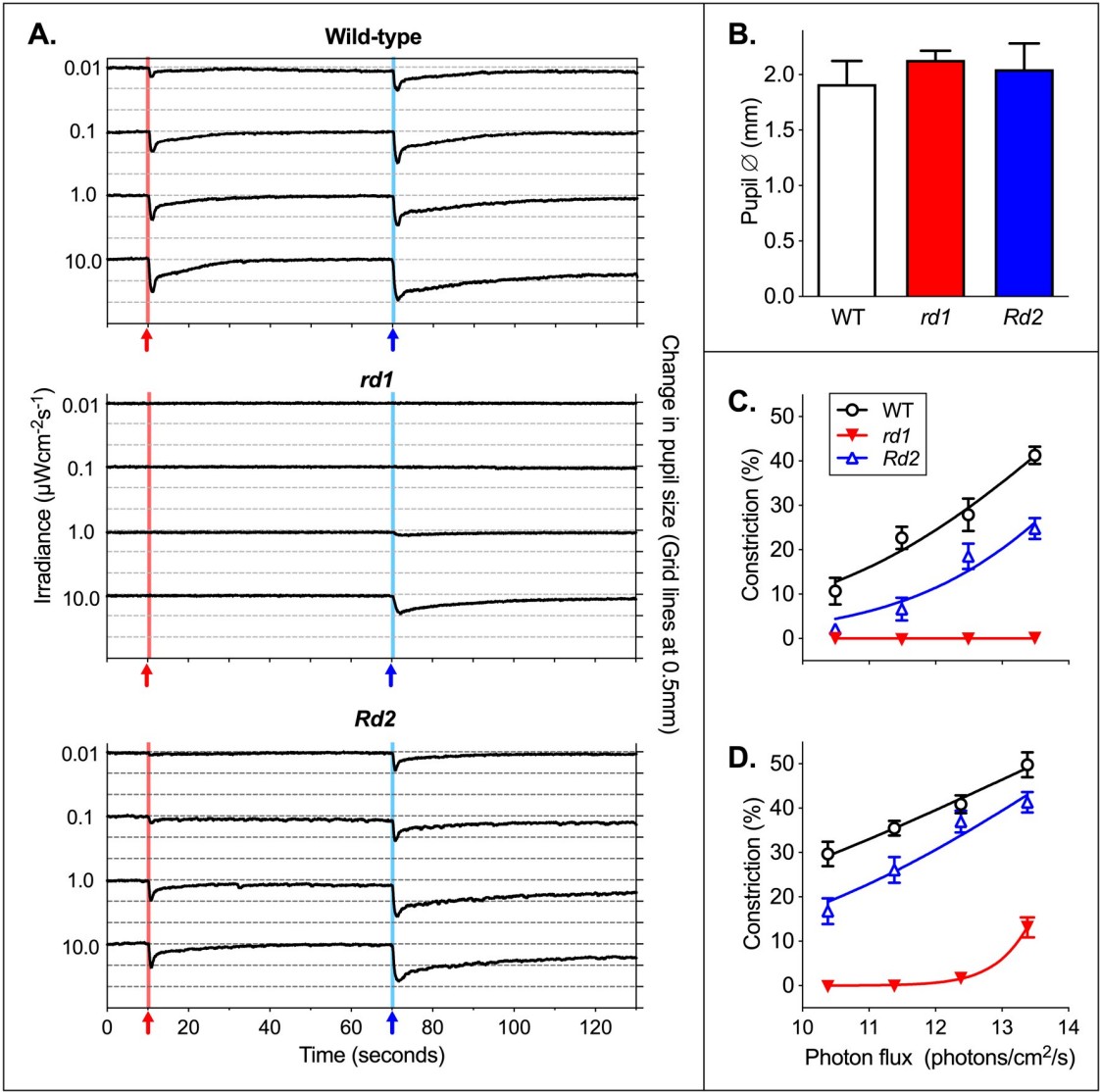

**Fig 4. Mean pupil traces at different irradiances.** (A) Combined mean traces of pupil size over time for wild-type (n = 9), *rd1* (n = 9), and *Rd2* mice (n = 10). Change in pupil size is shown in mm, with grid lines showing 0.5mm intervals. The 1-second red stimulus at 10-seconds is shown by a red arrow and red background line. The 1-second blue stimuli at 70 seconds is shown by a blue arrow and blue background line. (B) Dark adapted pupil diameter is shown in mm, with stimuli shown in power. Dark-adapted baseline pupil size was significantly larger in *rd1* (Mean ± SEM in mm: wild-type 1.92 ± 0.21; *rd1* 2.13 ± 0.08; P < 0.0001; t = 5.7; n = 9) and to a lesser degree in *Rd2* mice (2.05 ± 0.23; P = 0.014; t = 2.5; n = 9, 10). (C) Responses to a 1-second red stimulus at four irradiances are shown with dose response curves fitted. 622nm stimuli are shown in log10 photon flux: 10 $\mu Wcm^{-2}s^{-1}$ = 3.12x10$^{13}$ photons/cm²/s = 13.49 on a log10 scale. (D) Responses to a 1-second blue stimulus at four irradiances are shown with dose response curves fitted. 480nm stimuli are shown in log10 photon flux: 10 $\mu Wcm^{-2}s^{-1}$ = 2.42x10$^{13}$ photons/cm²/s = 13.38 on a log10 scale. (B, C and D) Mean ± SEM.

## Discussion

The goal of this study was to determine the capacity of the PLR and masking behavior for measuring retinal function in different forms of rod/cone photoreceptor degeneration in mice. Together our data show that in a given disease, two responses to light can be affected in opposite ways. For example, in *rd1* mice the amplitude of response to light is reduced for the PLR but increased for negative masking. Our data also show that for negative masking, different

types of rod/cone degeneration can have opposite effects on the change in irradiance generating a response: *rd1* responses were induced at lower irradiances than wild-type but *Rd2* responses required higher irradiances than wild-type. However, the change in sensitivity to light for the PLR, and for the positive masking response, did to some extent reflect the degree of loss in rod/cone function.

Negative masking in mice is an innate suppression of activity in brighter light, probably acting to reduce predation. In *rd1* mice, we observed a previously reported paradoxical increase in sensitivity, which therefore poorly represents the loss of rod/cone function [19, 37]. In *Rd2* mice, the reduction in negative masking sensitivity follows loss of rod/cone input, but underestimates the reduction in sensitivity of image-forming vision: 90-day-old *Rd2* mice require ~3 log units increased irradiance for useful vision, but the loss of sensitivity for negative masking is only 1.4 log units [8].

The PLR adjusts light entering the eye to optimize visual acuity. In *rd1* mice, the PLR was severely reduced, reflecting total loss of rod and cone photoreceptor cells. In *Rd2* mice, there was a smaller decrease in the PLR, but again, this underestimates the reduction in sensitivity of image-forming vision: loss of PLR sensitivity was less than 2 log units for red and blue stimuli [8]. The extent of the deficit in the *rd1* PLR means that even a limited rescue of rod/cone function might be measured by increased PLR amplitude, particularly with a red-light stimulus. Given the similar spectral sensitivity of melanopsin, rod-opsin and medium wavelength cone opsin among different mammals, and the effectiveness of red-light PLR testing in humans, red-light PLR should be effective in other mammalian species [10, 12, 13].

Positive masking is the effect of vision on running wheel use, similar to how we would move more quickly when we can see what is in front of us. In *rd1* mice, the absence of a positive masking response was consistent with the lack of useful vision [7]. In *Rd2* mice, the presence of positive masking only at relatively high light levels was consistent with useful vision from photoreceptor cells that are relatively insensitive to light [8]. Positive masking responses appear to have potential in both models of rod/cone degeneration. In *Rd2* mice, an effective rescue treatment would shift the threshold for positive masking to lower irradiances, and in *rd1* mice, it is possible that an effective rescue treatment would restore positive masking. However, negative and positive masking present differently in nocturnal and diurnal species, and in prey or predator species, therefore their application to other species will depend on the specific behaviors of other target species [24].

Rod/cone degeneration does lead to plastic changes in the retina [38, 39]. For example, rod to rod bipolar cell signaling is potentiated in rod photoreceptor degeneration, preserving voltage output and scotopic vision [40, 41]. Although speculative, it seems likely that the negative masking and PLR phenotypes, in part, reflect plastic changes in retinal cells and circuits. We have previously shown that the paradoxical increased sensitivity of negative masking seen in *rd1* mice, also emerges in very old *Rd2* mice after complete loss of rods and cones [42]. Further, melanopsin knockout mice have a reduced negative masking response, which implies rod/cone input is not inhibitory to melanopsin input for negative masking [43]. It follows that the simplest explanation for the paradoxical increased negative masking sensitivity, is that absence of rod/cone input causes a compensatory gain in this pathway, potentially at multiple levels in retinal and post-retinal circuits.

In summary, our data show that different diseases of the rods and cones can have contrasting effects on the non-image-forming responses to light that do not simply reflect rod/cone loss. Despite the divergent phenotypes and disparity between sensitivities for these responses and for image-forming-vision, it is our opinion that these non-image-forming responses to light can be useful as measures of retinal function in mice. First, consistency of measurement methods between pre-clinical and clinical tests can be particulalry compelling, and a PLR is

much more practical in young children than a dark-adapted ERG or Snellen chart [44]. Second, conditioned responses can be difficult to measure in some human subjects and animal models. For example, the success of RPE65 gene therapy was most clearly demonstrated by ability to negotiate mazes, but this required an equivalent distribution of learning and memory function among subjects, and useful vision for task learning [1, 3, 8, 45, 46]. However, it is essential that methods are carefully selected and a preliminary characterization made for effective testing in a given disease model.

## Supporting information

**S1 Checklist. The ARRIVE guidelines 2.0: Author checklist.**
(PDF)

## Author Contributions

**Conceptualization:** Stewart Thompson.

**Data curation:** Ethan O. Contreras, Carley G. Dearing, Stewart Thompson.

**Formal analysis:** Stewart Thompson.

**Funding acquisition:** Stewart Thompson.

**Investigation:** Ethan O. Contreras, Carley G. Dearing, Crystal A. Ashinhurst, Betty A. Fish, Sajila N. Hossain, Ariana M. Rey, Primal D. Silva, Stewart Thompson.

**Methodology:** Stewart Thompson.

**Resources:** Stewart Thompson.

**Supervision:** Stewart Thompson.

**Writing – original draft:** Stewart Thompson.

**Writing – review & editing:** Ethan O. Contreras, Stewart Thompson.

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
