## [Decision Letter · Decision Letter 0]

5 Jan 2021

PONE-D-20-39197

Pupil and masking responses to light as functional measures of retinal degeneration in mice Mus Musculus.

PLOS ONE

Dear Dr. Thompson,

Thank you for submitting your manuscript to PLOS ONE. After careful consideration, we feel that it has merit but does not fully meet PLOS ONE’s publication criteria as it currently stands. Therefore, we invite you to submit a revised version of the manuscript that addresses the points raised during the review process.

The reviewers were generally enthusiastic about your paper but offered useful suggestions about how to improve it.  These include changes to how you report your methods and results. In particular, please consider converting irradiance values to photon flux. For statistical purposes, please report an n value for each experiment.

We look forward to receiving your revised manuscript.

Kind regards,

Alfred S Lewin, Ph.D.

Academic Editor

PLOS ONE

Reviewers' comments:

Reviewer's Responses to Questions

**Comments to the Author**

1. Is the manuscript technically sound, and do the data support the conclusions?

Reviewer #1: Yes

Reviewer #2: Yes

2. Has the statistical analysis been performed appropriately and rigorously? 

Reviewer #1: Yes

Reviewer #2: Yes

3. Have the authors made all data underlying the findings in their manuscript fully available?

Reviewer #1: Yes

Reviewer #2: Yes

4. Is the manuscript presented in an intelligible fashion and written in standard English?

Reviewer #1: Yes

Reviewer #2: Yes

5. Review Comments to the Author

Reviewer #1: The manuscript proposed by Contreras et al. adds value to the vision science/ophthalmology field. The report is constructed and written in an intelligible fashion. Data representation is concise and clear. I have no major concerns regarding the experiments or data, but I have several minor issues that should be addressed before proceeding:

- I suggest to edit the main title as: Pupillary reflex and behavioral masking responses to light as functional measures of retinal degeneration in mice.

- I suggest removing Mus musculus from the title as redundant.

- In the abstract, I would suggest referring slower progression than slow in regards to Rd2 mice. It is still relatively fast disease compared to many other RD models.

- I am wondering why the authors write rd1 and Rd2 differently. I would say either both with capital, or then neither, for consistency.

- There is a recent review paper listing vision testing in rodents (Leinonen & Tanila, Behav Brain Res). It would be appropriate to refer to this paper after second sentence in introduction (currently citations 3-7)

- Looking at reference 7 makes me think that Rds mice at 3-month of age mostly only have cone function. They only have behavioral vision responses at mesopic/photopic light environments, and the ERG could completely arise from the cones. Therefore, I suggest to revise row 94-95 as "but at 3 months photoreceptors are responsive...". This will avoid the need to know whether it is rods or cones, or both, that respond.

- In row 116 authors say there is no end-points in this study. Do they mean, no humane endpoints?

- I am wondering why the authors used such high xylazine dose for the ERG. Typically it is 10 mg/kg. This drug has narrow therapeutic window.

- Row 139: cd.s/m2(supercript)

- Row 141-142: authors say they ran statistics for the ERG amplitude but no results are shown in figure (as e.g. asterisk) or in the figure legend.

- Row 162: please edit as "Genotype was masked for experimenter analyzing the data using non-identifying...". After that please remove "genotype group was identified..." sentence as non-necessary.

- Row 214: Same as earlier, maybe better to refer to photoreceptor function than rod and cone. I suspect that rods are not responding at all at that age.

- Fig. 1 legend: Please write ERG stimulus intensity here too and sample size.

- Same in all fig legends: Please give sample sizes.

- Please revise sentence in row 237-240. Most likely rods/cones have zero contribution to the masking behavior in Rd1 mice at that age.

- In rows 263-268, I suggest to highlight that constriction was absent for red stimulus in Rd1 mice, and prominently reduced for the blue stimulus. And in Rd2 mice, clearly reduced for red, but relatively less for the blue stimulus.

- Comma in row 281 is not needed.

- Authors should add results of statistical tests to Figure 4B (asterisks) and/or to the figure legend.

- Fig. 3: I suggest that authors add arrows to x-axis when red and blue stimulus starts.

- In the discussion, authors should highlight more that photoreceptor cell loss triggers compensatory changes downstream already in the level of bipolar cells. This was very recently shown both with an induced photoreceptor ablation model (Care...Dunn 2020, Cell Reports) and in progressive retinitis pigmentosa (Leinonen et. al. 2020, eLife) in mice. As this happens already in the first visual synapse, it is no surprising that sensory loss does not directly translate to visual behavior. More compensation likely occur in the level of ganglion cells, and perhaps further in the brain circuits.

- I missed the explanation why studying non-image forming vision is important from the therapy point of view? If a therapy cannot reproduce, rescue or prevent the loss of image-forming vision, can it anyhow be considered successful therapy? Could authors comment on this in the discussion.

Reviewer #2: Testing visual capabilities in mouse models of retinal degeneration is a recurrent issue in studies aimed at understanding ocular diseases and developing treatments. Assays for spatiotemporal acuity, such as ERG, optokinetic / optomotor reflexes or visual water maze are typically ineffective in measuring low levels of functional vision. The authors propose an alternative strategy based on non-image forming behaviours, such as pupil-light reflex (PLR) and enhancement/suppression of locomotor activity (wheel running), that correlate with visual acuity but are expressed robustly even in animals with very low functional vision.

The study is well performed, the experimental methods are described in sufficient details and the statistical analyses, that include a blinding strategy, are competently reported. I only have one major comment and few minor suggestions (see below).

Major Comments:

Irradiance values, in μW/cm2s-1 are somewhat hard to interpret in terms of the underlying rod/cone/melanopsin physiology. Converting those values to photon flux would facilitate comparison with previous works, such as e.g. response (e.g. melanopsin driven PLR, PMID:11369943, Fig.3). This is even more important for blue and red light where definition of blue and red varies widely for different light sources. An estimate of photon flux for rod, S-cones, M-cones and melanopsin would also be important for replicating the authors’ results.

Minor Comments:

For locomotion data an additional panel showing the raw amount of wheel running for each genotype would provide useful information about baseline behaviour.

Line 67: Of course diffuse bright light suppresses locomotion over long epochs and this has been widely reported. However, over short epochs (0-1seconds), an equally robust effect in increasing locomotion in wild-type and rd1 mice (PMID:31316114) and other types of motor activity (e.g. rearing, PMID:33007242) has also been recently reported. This could be added to the list of non-image forming responses.

Line 102: The authors are making a great contribution to the scientific community by sharing their dataset. However the link (doi:10.5061/dryad.9kd51c5g5) appears to be broken when I tried it.

Figure 3: Although this information is provided in caption, blue and red background lines at 10 and 70s, like in Fig.4, would help the reader.

Statistics: the n number for each statistics should be reported

6. PLOS authors have the option to publish the peer review history of their article (what does this mean?). If published, this will include your full peer review and any attached files.

Reviewer #1: No

Reviewer #2: No

---

## [Author Response · Author response to Decision Letter 0]

8 Jan 2021

Response to Review Comments

Reviewer 1.

I suggest to edit the main title as: Pupillary reflex and behavioral masking responses to light as functional measures of retinal degeneration in mice.

• This edit does clarify the subject of the paper, and we appreciate the suggestion.

I suggest removing Mus musculus from the title as redundant.

• We agree with the reviewer on this point but added Mus Musculus because PLOS One guidelines state that formal species name be given in the title. We have removed the redundant species name but will reinstate it if required by PLOS One

In the abstract, I would suggest referring slower progression than slow in regards to Rd2 mice. It is still relatively fast disease compared to many other RD models.

• We appreciate this suggestion for a more accurate description, and have made this change.

I am wondering why the authors write rd1 and Rd2 differently. I would say either both with capital, or then neither, for consistency. 

• For rd1 the lower case refers to recessive disease allele, while upper case in Rd2 refers to a dominant disease mutation, so we have followed this principle of correct nomenclature for mice. Although both disease models were homozygous for their respective disease alleles, mice heterozygous for the Rd2 allele do have some structural and functional deficits.

There is a recent review paper listing vision testing in rodents (Leinonen & Tanila, Behav Brain Res). It would be appropriate to refer to this paper after second sentence in introduction (currently citations 3-7).

• Thank you for the direct to this nice review paper. Citation added.

Looking at reference 7 makes me think that Rds mice at 3-month of age mostly only have cone function. They only have behavioral vision responses at mesopic/photopic light environments, and the ERG could completely arise from the cones. Therefore, I suggest to revise row 94-95 as "but at 3 months photoreceptors are responsive...". This will avoid the need to know whether it is rods or cones, or both, that respond.

• It is true that we do not know if rods, cones or both underlie the vision in relatively bright light levels. The presence of vision in mesopic/photopic light levels could arise from rods because they only have 0.1% of the opsin of normal rods, so will need relatively high light levels for threshold photon capture, but this is not certain so the suggested edit is more correct. 

In row 116 authors say there is no end-points in this study. Do they mean, no humane endpoints?

• Yes. Thank you for the clarifying edit suggestion.

I am wondering why the authors used such high xylazine dose for the ERG. Typically it is 10 mg/kg. This drug has narrow therapeutic window.

• Typo. Standard Ratliff mouse mix was used. Edit made.

Row 139: cd.s/m2(supercript)

• Typo. Edit made.

Row 141-142: authors say they ran statistics for the ERG amplitude but no results are shown in figure (as e.g. asterisk) or in the figure legend.

• Now included in the appropriate point of Fig1 legend as “Reduction in ERG b-wave was significant for both rd1 (Mann-Whitney test P < 0.001) and Rd2 mice (P < 0.001).”

Row 162: please edit as "Genotype was masked for experimenter analyzing the data using non-identifying...". After that please remove "genotype group was identified..." sentence as non-necessary.

• Edit made.

Row 214: Same as earlier, maybe better to refer to photoreceptor function than rod and cone. I suspect that rods are not responding at all at that age.

• Edit made.

Fig. 1 legend: Please write ERG stimulus intensity here too and sample size.

Same in all fig legends: Please give sample sizes.

• Edits made.

Please revise sentence in row 237-240. Most likely rods/cones have zero contribution to the masking behavior in Rd1 mice at that age.

• Now reads “suggesting a change in the quantum efficiency: this would be consistent with a change in photoreceptor contribution to the response with the complete loss of rod and cone photoreceptors.”

In rows 263-268, I suggest to highlight that constriction was absent for red stimulus in Rd1 mice, and prominently reduced for the blue stimulus. And in Rd2 mice, clearly reduced for red, but relatively less for the blue stimulus.

• Now reads “However, in rd1 mice there was no pupil constriction to bright red light (10 μWcm-2s-1 at 622nm) and a pronounced reduction in the response to bright blue light (10μWcm-2s-1 at 480nm). In Rd2 mice there was pupil constriction to bright red and bright blue light but amplitude was reduced for both stimuli, and with a more pronounced loss of response amplitude to red light than to blue light.”

Comma in row 281 is not needed.

• Edits made.

Authors should add results of statistical tests to Figure 4B (asterisks) and/or to the figure legend. 

• Statistics moved from results paragraph to figure legend.

Fig. 3: I suggest that authors add arrows to x-axis when red and blue stimulus starts.

• Edit made in figure and legend.

In the discussion, authors should highlight more that photoreceptor cell loss triggers compensatory changes downstream already in the level of bipolar cells. This was very recently shown both with an induced photoreceptor ablation model (Care...Dunn 2020, Cell Reports) and in progressive retinitis pigmentosa (Leinonen et. al. 2020, eLife) in mice. As this happens already in the first visual synapse, it is no surprising that sensory loss does not directly translate to visual behavior. More compensation likely occur in the level of ganglion cells, and perhaps further in the brain circuits.

• Given this valid prompt, the discussion has been altered to better convey the evidence for compensation in signal amplitude with photoreceptor degeneration. Penultimate paragraph of the discussion now reads: Rod/cone degeneration does lead to plastic changes in the retina [35,36]. For example, rod to rod bipolar cell signaling is potentiated in rod photoreceptor degeneration, preserving voltage output and scotopic vision [37,38]. Although speculative, it seems likely that the negative masking and PLR phenotypes, in part, reflect plastic changes in retinal cells and circuits. We have previously shown that the paradoxical increased sensitivity of negative masking seen in rd1 mice, also emerges in very old Rd2 mice after complete loss of rods and cones [39]. Further, melanopsin knockout mice have a reduced negative masking response, which implies rod/cone input is not inhibitory to melanopsin input for negative masking [40]. It follows that the simplest explanation for the paradoxical increased negative masking sensitivity, is that absence of rod/cone input causes a compensatory gain in this pathway, potentially at multiple levels in retinal and post-retinal circuits.”

I missed the explanation why studying non-image forming vision is important from the therapy point of view? If a therapy cannot reproduce, rescue or prevent the loss of image-forming vision, can it anyhow be considered successful therapy? Could authors comment on this in the discussion.

• Given this valid prompt, the discussion has been altered to better convey the potential value of these tests. Last paragraph of the discussion now reads: “In summary, our data show that different diseases of the rods and cones can have contrasting effects on the non-image-forming responses to light that do not simply reflect rod/cone loss. Despite the divergent phenotypes and disparity between sensitivities for these responses and for image-forming-vision, it is our opinion that these non-image-forming responses to light can be useful as measures of retinal function in mice. First, consistency of measurement methods between pre-clinical and clinical tests can be particulalry compelling, and a PLR is much more practical in young children than a dark-adapted ERG or Snellen chart [38]. Second, conditioned responses can be difficult to measure in some human subjects and animal models. For example, the success of RPE65 gene therapy was most clearly demonstrated by ability to negotiate mazes, but this required an equivalent distribution of learning and memory function among subjects, and useful vision for task learning [1,3,8,39,40]. However, it is essential that methods are carefully selected and a preliminary characterization made for effective testing in a given disease model.”

Reviewer 2.

Major Comments:

Irradiance values, in μW/cm2s-1 are somewhat hard to interpret in terms of the underlying rod/cone/melanopsin physiology. Converting those values to photon flux would facilitate comparison with previous works, such as e.g. response (e.g. melanopsin driven PLR, PMID:11369943, Fig.3). This is even more important for blue and red light where definition of blue and red varies widely for different light sources. An estimate of photon flux for rod, S-cones, M-cones and melanopsin would also be important for replicating the authors’ results.

• We have changed figure 4 to show photon flux where appropriate and have included relative quantum efficiency for SWS-cone opsin, melanopsin, rod-opsin and MWS-cone opsin. The section for the PLR in the methods now reads: Pupillary light reflexes were tested using a chromatic contrast approach as previously described [31]. This approach exploits differences in efficiency of photopigment activation at different wavelengths. Opsin photopigments have a characteristic absorption spectrum but the wavelength that most efficiently activates a mouse photopigment (max) is specific to that pigment: Short-Wavelength-Sensitive (SWS) opsin = 360nm, Melanopsin = 480nm, Rod-opsin = 498nm, and Medium-Wavelength-Sensitive (MWS) opsin = 508nm. Relative quantum efficiency for mouse photopigments corrected for lens-absorption then normalized to 1 is: 480nm SWS = 0.00001, Melanopsin = 1.0, Rod-opsin = 0.92, and MWS = 0.83; 622nm SWS <0.00001, Melanopsin = 0.0006, Rod-opsin = 0.005, and MWS = 0.013 [32]. Although the percentage of photons that will activate an opsin is much lower for all mouse photopigments at 622nm, it is higher for Rod-opsin (0.5%), and MWS-opsin (1.6%) than it is for Melanopsin (0.06%). This means the 622nm PLR can be largely attributed to the rod and MWS-cone. The chromatic PLR was measured in wild-type (n = 9, 4 male, 5 female), Rd2 (n = 10, 5 male, 5 female), and rd1 (n = 9, 4 male, 5 female) mice according to previously described protocols [31]. 

Minor Comments:

For locomotion data an additional panel showing the raw amount of wheel running for each genotype would provide useful information about baseline behaviour.

• Added as suggested. Section in results now includes: Baseline wheel running activtiy was not different between wild-type (wheel revolutions between Zeitgeber time 14 and 15, Mean ± SD = 294.0 ± 25.1) and rd1 mice (Mean ± SD = 274.3 ± 64.5; Two tailed unequal variance t-test P = 0.33; n = 12), or between wild-type and Rd2 mice (Mean ± SD = 311.4 ± 79.0; P = 0.48; n = 12). 

Line 67: Of course diffuse bright light suppresses locomotion over long epochs and this has been widely reported. However, over short epochs (0-1seconds), an equally robust effect in increasing locomotion in wild-type and rd1 mice (PMID:31316114) and other types of motor activity (e.g. rearing, PMID:33007242) has also been recently reported. This could be added to the list of non-image forming responses.

• Added as suggested.

Line 102: The authors are making a great contribution to the scientific community by sharing their dataset. However the link (doi:10.5061/dryad.9kd51c5g5) appears to be broken when I tried it.

• We used a PLOS ONE recommended data depository, but upload was recent and is now working.

Figure 3: Although this information is provided in caption, blue and red background lines at 10 and 70s, like in Fig.4, would help the reader.

• Added as suggested.

Statistics: the n number for each statistics should be reported.

• Added as suggested.

---

## [Editor Report · Decision Letter 1]

11 Jan 2021

Pupillary reflex and behavioral masking responses to light as functional measures of retinal degeneration in mice.

PONE-D-20-39197R1

Dear Dr. Thompson,

We’re pleased to inform you that your manuscript has been judged scientifically suitable for publication and will be formally accepted for publication once it meets all outstanding technical requirements.

Kind regards,

Alfred S Lewin, Ph.D.

Section Editor

PLOS ONE
---

## [Editor Report · Acceptance letter]

14 Jan 2021

PONE-D-20-39197R1 

Pupillary reflex and behavioral masking responses to light as functional measures of retinal degeneration in mice. 

Dear Dr. Thompson:

I'm pleased to inform you that your manuscript has been deemed suitable for publication in PLOS ONE. Congratulations! Your manuscript is now with our production department. 

Kind regards, 

on behalf of

Dr. Alfred S Lewin 

Section Editor

PLOS ONE